# Physical Inactivity Levels of European Adolescents in 2002, 2005, 2013, and 2017

**DOI:** 10.3390/ijerph20043758

**Published:** 2023-02-20

**Authors:** Jorge López-Fernández, Alejandro López-Valenciano, Gemma Pearce, Robert J. Copeland, Gary Liguori, Alfonso Jiménez, Xian Mayo

**Affiliations:** 1Faculty of Sport Sciences, Universidad Europea de Madrid, 28670 Madrid, Spain; 2GO fit LAB, Ingesport, 28003 Madrid, Spain; 3Department of Education Science, Universidad Cardenal Herrera-CEU, CEU Universities, 12006 Castellón de la Plana, Spain; 4Centre for Healthcare Research, Coventry University, Coventry CV1 5FB, UK; 5Advanced Wellbeing Research Centre, Sheffield Hallam University, Sheffield S1 1WB, UK; 6The National Centre for Sport and Exercise Medicine, Sheffield S9 3TY, UK; 7Department of Movement Sciences, University of West Florida, Pensacola, FL 32514, USA; 8Observatory of Healthy & Active Living of Spain Active Foundation, Centre for Sport Studies, King Juan Carlos University, 28942 Madrid, Spain

**Keywords:** active behaviour, MVPA, national policies, physical activity, youth

## Abstract

Sport and Physical Activity (PA) Special Eurobarometer surveys may inform of the physical inactivity (PIA) levels in the European Union (EU). This study aimed to analyse the PIA levels of EU adolescents (15–17 years) in four time points, according to gender. The data were from 2002, 2005, 20013, and 2017 Special Eurobarometers. Adolescents were categorised as “Inactive” when performing less than 60 min/day of moderate to vigorous PA on average. A χ^2^ test was used to compare the levels of PIA between survey years. PIA levels between gender were analysed using a Z-score test for two population proportions. PIA levels ranged from 67.2% for boys (59.4% to 71.5%;) to 76.8% for girls (76.0% to 83.4) across the time points. Adjusted standardised residuals revealed a decrease in the observed levels versus the expected for 2005 (whole sample: −4.2; boys: −3.3) and an increase for 2013 (whole sample: +2.9; boys: +2.5). Boys presented lower PIA levels than girls in all years (*p* ≤ 0.003), but descriptively, the difference progressively decreased (from 18.4% to 11.8%). No significant reductions in PIA levels were observed between 2002 and 2017, and girls reported consistently higher levels of PIA than boys.

## 1. Introduction

Physical activity (PA) enhances fitness, bone, and muscular health while associated with psychological and physiological benefits in childhood and adulthood [1,2,3]. On the contrary, the default of meeting Global Recommendations on physical activity, defined as physical inactivity (PIA), is associated with a negative impact on adolescents’ physical, mental, and social health while increasing the risk of suffering non-communicable diseases during adulthood. [1,2,3]. Hence, enhancing PA and addressing PIA is identified as a public health priority, and the European Commission supports the World Health Organization’s (WHO) global objective of reducing the PIA levels at a population level by 15% by 2030 [3]. To assist European countries in achieving this objective, population-based action plans based on 23 factors that can positively influence health-enhancing physical activity have been developed [4].

Among all population groups, children and adolescents from 5 to 17 years are especially important to target in this initiative, given that weekly MVPA declines during the transition from adolescence to adulthood [5]. In addition, the PIA behaviour adopted during adolescence is likely to be maintained during adulthood [6]. At the same time, adults who were inactive or had low cardiovascular fitness during childhood are more likely to develop health conditions in later life [6]. Moreover, PIA in adolescents and children is associated with paediatric dynapenia, reduced fundamental movement skills, osteopenia, and cardiometabolic disorders [7,8,9]. Based on the Global Recommendations on Physical Activity (PA), to achieve the health-related benefits of physical activity, children and adolescents should perform at least 60 min/per day of moderate-to-vigorous intensity aerobic PA (MVPA) across the week. In addition, exercises that strengthen muscles and bones should be undertaken at least three days a week [1]. Adolescents that do not meet these global recommendations are classified as physically inactive [1].

Up to 2020, the existing estimations for Europe suggested that at least 75% of adolescents (11–17 years old) did not meet the global recommendations for PA regardless of the European country [10,11], though great variation exists for PIA levels among countries (i.e., 40–50% PIA for England, Lithuania, and the Netherlands; 20% PIA for Slovenia) [11]. Moreover, the PIA level was higher among girls than boys [10], so further efforts to decrease PIA and/or increase PA among adolescents is a prudent public health practice.

Initiatives from European research networks such as The Global Matrix Report Card [11,12] and the Health Behaviour in School-aged Children (HBSC) [13,14] or the work from Guthold et al. [10] and Steene-Johannessen et al. [15] provide data about PA levels amongst adolescents across the European Union (EU). However, these reports come from private, not-for-profit initiatives that were not part of the European Union surveillance system on physical activity and had some limitations. For instance, The Global Matrix 3.0 Report Card [11] did not provide data for girls and boys separately, nor did it include all European Union countries. This is partially overcome in the latest report (Global Matrix Report Card 4.0.), which reports PA prevalence in several countries around 2022 considering the gender (boys and girls) or where they are living (i.e., rural or urban), but still, these data do not cover the whole European Union country and many European countries failed in providing separate data for girls and boys [16]. Moreover, changes through time have been analysed using the Global Matrix Report Card from 1.0. to 4.0. still, data for the European Union is not provided [17]. The HBSC project compared two time points and separate data for boys and girls, but weekly MVPA was measured via a single-item self-reported questionnaire [13,14]. Guthold et al. [10] reported PA among many countries, including Europe, but they used different datasets with different methodologies to report PA. Finally, Steene-Johannessen et al. [15] measured the PA levels of children and adolescents in 18 European Countries through accelerometery. However, not all European countries participated in the study or analysed changes over time. Furthermore, none of these networks and studies provide specific analysis for the EU as a territory at different time points.

To understand the impact of European public health initiatives to increase PA, further research is needed to know how PIA levels have varied among EU adolescents over time. Special Eurobarometer surveys conducted periodically by the European Commission are part of the European Surveillance System of PA from the European Commission, so is a primary dataset that can be used to analyse the PIA levels of the EU population (combined and by gender) across four time points (2002, 2005, 2013, and 2017) [18]. Therefore, as happened with European adults [18], this data should help evaluate and monitor the HEPA Monitoring Framework of the European Commission within the studied period [4,19] and set baseline data for PIA among the European Union as part of the 2018 WHO Global Action Plan framework [3]. However, to date, no studies have used this data to explore the PIA levels of EU adolescents across different time points despite these data being part of the European Commission surveillance system on PA. Therefore, the primary aim of this study was to a) analyse the levels of PIA of EU adolescents across four different time points (2002, 2005, 2013, and 2017); and b) compare the PIA levels according to gender.

## 2. Materials and Methods

### 2.1. Design

Descriptive epidemiology study. The research has been checked against the STROBE reporting guidelines (Appendix A).

### 2.2. Data Source

For this study, data from 15–17-year-old adolescents were obtained from four successive Special Eurobarometer surveys that recorded data on PA and health from EU citizens: December 2002 (Special Eurobarometer 183.6; *n* = 543), December 2005 (Special Eurobarometer 246; *n* = 929), December 2013 (Special Eurobarometer 412; *n* = 592), and December 2017 (Special Eurobarometer 472; *n* = 478). The total sample included 2542 adolescents (1207 girls and 1335 boys) from the 28 EU member countries (Austria, Belgium, Bulgaria, Czech Republic, Croatia, Cyprus Republic, Denmark, Estonia, Finland, France, Germany [combined West and East Deutschland], Great Britain, Greece, Hungary, Ireland, Italy, Latvia, Lithuania, Luxembourg, Malta, Netherlands, Poland, Portugal, Romania, Slovakia, Slovenia, Spain, and Sweden). Northern Cyprus and Turkey were not analysed as they do not belong to the EU member countries. Following the inclusion criteria used in previous studies using Eurobarometer data, Northern Ireland was not considered [18]. As with other Eurobarometer surveys, these surveys were conducted using a multi-stage sampling, random design. Accordingly, to cover the country’s whole territory, the number of sampling points was drawn with probability proportional to both population size and population density.

### 2.3. Measures

The PIA levels of European adolescents were determined using the International Physical Activity Questionnaire (IPAQ-SF) [20] included in the selected Eurobarometer surveys. This questionnaire has been designed to measure the intensity (walking, moderate and vigorous), frequency, and duration of the PA performed by individuals aged 15–69 [20,21]. Although the convergent validity of the IPAQ questionnaires is limited [22], it has been recently used in different studies with European adolescents [23]. Furthermore, data from Eurobarometer surveys have already been used to determine the PIA levels of European adults [18]. Adolescents participating in the Eurobarometer surveys were asked about the number of days and amount of vigorous, moderate, and walking PA they participated in. The 2002 and 2005 surveys used the classical open solution for duration, so no specific responses were provided. This was changed in the 2013 and 2017 surveys, which truncated five different fixed possibilities (0 min; 1 to 30 min; 31 to 60 min; 61 to 90 min; 91 to 120 min; more than 120 min).

To reduce bias from the different approaches between databases, responses from the surveys conducted in 2002 and 2005 were truncated according to the methodology used in the 2013 and 2017 surveys. Thus, for the case of PA, responses of “30 min or less” were assumed to mean 15 min; “31 to 60 min” was assumed to mean 45 min; “61 to 90 min” was assumed to mean 75 min; “91 to 120 min” was assumed to mean 105 min; and “more than 120 min” was assumed to mean 120 min [18]. The data processing and analysis were completed using a modified ad hoc spreadsheet available online [24] according to the instruction for data processing and analysis of the IPAQ-SF [21] and the methodology used in recent studies [18]. Following the recommendations of previous studies using the IPAQ-SF in adolescents, walking activities were not included in the analysis [25]. Adolescents were categorised as “inactive” (Performing <60 min/day of MVPA on average) or “active” (performing ≥60 min/day of MVPA on average) [1]. Only individuals with a valid response (i.e., different answer than “don’t know” or “error”) for intensity and duration of a particular intensity (i.e., moderate or vigorous PA) were analysed [25].

### 2.4. Statistical Analysis

Descriptive statistics, presented as a proportion (%) with a 95% confidence interval (95% CI), were calculated for the inactive/active dichotomic variable. The χ^2^ test was implemented to study the association between PA level (inactive and active) and time points for the whole sample and boys and girls. Due to the number of EU countries increasing from 15 to 28 in 2004, two analyses were performed comparing the four time points. The first analysis considered data from all countries participating in each Special Eurobarometer. The second analysis compared the data from the first 15 countries entering the EU before 2004. In both cases, the analysis of the adjusted standardised residuals was conducted when a significant association was found. A Z-score for two population proportions was used to identify differences by gender (girls vs. boys). An a priori alpha level was set at 0.05. Z-score analyses were performed with Microsoft Excel version 1709 (Microsoft Corporation; Redmond, Washington, United States of America). The remaining analyses were performed using the Statistical Package for Social Sciences (version 22.0, SPSS Inc., Chicago, IL, USA). Comparative analysis among EU countries was not performed due to the low sample size for each country in all studied time points (Supplementary File S2 shows data from each EU country).

## 3. Results

The descriptive outcomes for PIA levels of EU adolescents in each of the four time points studied are displayed in Table 1. Data are provided for the total sample and boys and girls. The PIA levels of the four time points ranged from 67.2% (64.1–70.5%) to 76.8% (73.0–80.5%). The comparison of PIA levels among the studied time points revealed significant differences for the whole sample (*n* = 2414; χ2 = 22,461; DF = 3; *p* < 0.001; and for boys (*n* = 1282; χ2 = 16,713; DF = 3; *p* = 0.001), but not for girls (*n* = 1132; χ2 = 7796; DF = 3; *p* = 0.05) (Figure 1). In the whole sample and boys, the analysis of the standardised residuals showed a decrease in the levels of PIA observed versus the expected for 2005 (whole sample: −4.2; boys: −3.3) and an increase in the levels observed versus expected for 2013 (whole sample: +2.9; boys: +2.5). When the outcomes considered only the first 15 EU countries, there were no differences in PIA levels for the whole sample (n = 1411; χ2 = 5154; DF = 3; *p* = 0.161) or for boys (*n* = 744; χ2 = 3668; DF = 3; *p* = 0.300) and girls (*n* = 667; χ2 = 2052; DF = 3; *p* = 0.562), separately. Boys showed lower PIA levels in all time points than girls (2002 [−18.4 percentual points; *p* < 0.001; Z-score = 4.359]; 2005 [−16.6 percentual points; *p* < 0.001; Z-score = 5.383]; 2013 [−12.5 percentual points; *p* < 0.001; Z-score = 3.489]; 2017 [−11.8%; *p* = 0.003; Z-score = 2.946]).

## 4. Discussion

PA in adolescents’ physical, social, and psychological health is widely accepted [1,2]. However, data suggests that large proportions of this population across the globe do not meet global recommendations [10,11,12,13,14,15]. The 2018 WHO Global Action Plan [3] and numerous country-level public health strategies attempt to address this lack of activity. To help understand the impact of policy and practice in the EU by the time the 2018 Global Action Plan was published, this study analysed the levels of inactivity amongst EU adolescents between 2002 and 2017 using four different time points. This study revealed (a) adolescents reported high PIA levels regardless of the studied year; (b) an increase in PIA levels occurred between 2002 and 2017; (c) girls have higher PIA levels than boys regardless of the cohort; and d) the difference between boys and girls progressively reduced over time.

Previous research has assessed the PIA levels of children and young adults in multiple countries, including EU countries, before and coinciding with the publication of the 2018 Global Action Plan, with results comparable to those reported here (PIA levels of 76.8% in our studies to ~80% in previous studies) [2,10,11]. However, the present study is the first to use the data from the Eurobarometer surveys conducted by the European Commission to analyse the PIA levels of European adolescents considering gender (i.e., boys and girls) across four different time points (2002, 2007, 2013, and 2017). It is also the first study reporting data for the EU region using data from the European Commission surveillance system. Therefore, it might help to understand how effective European policies were in enhancing physical activity between 2002 and 2017.

In line with previous studies, PIA levels of EU adolescents before 2020 were high, and no improvements were achieved between 2002 and 2017. For instance, studies derived from the HBSC project did not find significant changes in the PIA levels of adolescents between 2006 and 2014 and between 2014 and 2018. Still, changes in PIA levels of specific EU countries were reported [13,14]. Furthermore, Guthold et al. [10] reported a significant increase in the PIA levels of boys between 2001 and 2016, which aligns with the increase in PIA levels of boys between 2005 and 2013 observed here. Our research also corroborates that, contrary to the period between 1988 and 2002 in which PIA increased [26], PIA levels of EU adolescents between 2002 and 2018 remained relatively stable. The stable nature of PIA levels in our study years seems to have occurred despite the WHO’s 2012–2018 actions to reduce PIA [27], indicating that more needs to be done to address PIA amongst adolescents. Although data cannot be comparable as different methodologies and sample sizes are used, the latest data on PA among European adolescents (21 European countries) suggest some improvements have been achieved since 2017, but further effort is needed as still 61% to 66% of European adolescents report not meeting the global recommendations for PA [12].

Some reasons for the high PIA levels among adolescents reported in the studied period are that adolescents spend up to 70% of their waking time in sedentary activities (around 9 h/day) [28]. This sedentary activity is, on average, made up of after-school time (from 27.7% to 88.9%) [29], leaving little time for PA. Additionally, the daily sitting time of European adolescents remained steady throughout the years studied [30], indicating no increase in time available for engaging in PA. Accordingly, promoting plans to substitute sedentary activities for PA or increasing the opportunities to participate in PA across different contexts (at secondary school, leisure time, etc.) should be considered a public health priority [31].

The findings from the present research show the existence of a gender gap in all studied time points, although there is evidence of a progressive reduction in the gap. The decrease in the PIA gap is not from any reductions in PIA but instead appears to be due to a progressive increase in the PIA levels of boys. In contrast, girls’ PIA levels remain relatively similar, which has been reported elsewhere [10,13,14]. In accordance with our study, most existing research shows a gender gap in PA and PIA, with boys showing lower PIA levels and higher engagement in vigorous PA than girls [13,14]. Furthermore, this gap is well documented to continue into adulthood [18]. The gender gap in PA was already acknowledged by the European Commission in 2014 when developing the Gender Equality in Sport Proposal for Strategic Actions 2014–2020, setting the reduction of this gap as a priority for national and international sports organisations and policymakers [32]. Addressing gender stereotypes [33] and encouraging girls to participate in PA during playtime, in classes, and after school may positively reduce this gap [32]. However, having a clear understanding of the gender differences in leisure activities, particularly after school, may contribute to developing effective actions to reduce gender gaps [32].

In the last decade, the European Commission and the WHO Regional Office for Europe have encouraged the EU countries to implement national recommendations on PA, creating a monitoring and surveillance structure to periodically register the PIA levels, improving available time for physical education classes, training physical education teachers to deliver health-enhancing PA, promoting active breaks at school, and commuting to school [12,19,27,32]. However, the lack of improvements in PIA may be partly due to a miss-implementation of certain WHO actions. For instance, insufficient enhancement of active breaks at high schools or active travel, as was observed by half or more of the EU countries in 2018 [31]. In addition, physical education classes represented only two hours per week in most EU countries’ curricula and were not mandatory in secondary schools of seven countries [34]. Moreover, EU countries have not made public the outcomes of their country’s implemented actions/plans, only the existence of plans. Thus, the depth and development of the reported actions may have been unclear, including their efficacy on PA promotion among adolescents [35,36]. Consequently, further efforts from EU policymakers are important in order to enhance PA among adolescents and effectively address the gender gap in PA. This is particularly important given that adolescence is critical for adopting healthy PA behaviours and other health habits that continue in adulthood [6].

The present study also evidences that despite encouraging EU countries to monitor the PA of children and adolescents over time, the monitoring structure of the European Commission up to 2017 is not fully capturing health behaviours in children and adolescents. For instance, (i) contrary to adults [18], neither adolescents’ PIA levels of a particular EU country nor intercountry benchmarking can be set due to the low sample size from each country (<50 responses per country/year), (ii) Eurobarometer surveys do not provide data regarding the PA behaviour of children below 15 years old, nor how PA is accrued amongst this population (e.g., at or outside the educative centre). This lack of data makes it difficult to accurately monitor the PA habits of children and adolescents despite it being one of the main policy actions promoted by both the European Commission and the WHO [3,25], and 26 of the 28 EU countries implemented a health-behaviour surveillance system in 2018 [25]. Furthermore, (iii) these surveys do not cover other populations (e.g., children and adolescents with disability), while the low sample size does not permit the analysis of PA differences between rural and urban areas [16]. Moreover, (iv) the instrument used to measure PA among EU countries is not harmonised with other worldwide initiatives [36], and the validity of the IPAQ is questioned [22,36]. Finally, (v) changes in methodology were performed after 2013, which might lead to bias.

Based on the findings of this work and aligned with previous works [12,36], the authors of this study encourage European guideline developers and policymakers to improve monitoring and surveillance by (i) increasing representation of adolescents by country to allow both benchmark comparisons among European countries and strengthen the comparison analysis between girls and boys; (ii) extending the survey to other child populations (i.e., pubertal, prepuberal, infants, children with disability, etc.) [16]; (iii) monitoring the PA pattern either at or out of the educative centre; (iv) harmonising methodology in order to facilitate [36]. We also acknowledge that the Global Matrix Report Card [12,16] and other initiatives [13,14,15] can assist the European Commission in setting up a more comprehensive surveillance system. For instance, by analysing the PA behaviour of children and adolescents from rural and urban areas and in different contexts (e.g., physical education, active play, organised sports, structured PA, active transportation, etc.) [12], or by providing accelerometery data [15]. However, they are not-for-profit initiatives that do not depend on the European Commission, do not cover all EU countries, and do not follow a multi-stage sampling, random design to cover the whole territory of each participating country; moreover, to understand changes in PA from adolescence to adulthood comparative data is needed, which is not the case with existing approaches.

This study has some limitations to be acknowledged: (a) the sample size from each of the EU countries analysed using the Eurobarometer data is low, so findings should be interpreted cautiously; (b) PIA levels were measured by using the vigorous and moderate items of the IPAQ-SF, which is self-reported data with known shortfalls, and it does not inform of the full spectrum of PA behaviour among adolescents (e.g., physical education, active play, organised sports, and structured PA, active transportation, etc.) [12,36]. Although, as suggested in previous studies, despite these limitations, the use of these data for public health purposes should be valid as it does not apply to individuals but to group or year comparisons [30], and it is the only data available from the European Commission and all European Union Countries in different periods; (c) the IPAQ-SF used in 2002 and 2005 surveys had the classic open solution for minutes in both vigorous and moderate PA. At the same time, the 2013 and 2017 surveys had the possible answers for minutes truncated to several categorical response options. This might bias the comparison among years.

## 5. Conclusions

European adolescents show very high levels of PIA regardless of their gender. No improvements in PIA levels were observed between 2002 to 2017. Boys show lower PIA levels than girls in all studied years, although the gender gap has decreased descriptively between 2002 and 2017. The lack of improvements in PIA levels supports the necessity of further efforts to enhance active behaviour among adolescents.

## Figures and Tables

**Figure 1 ijerph-20-03758-f001:**
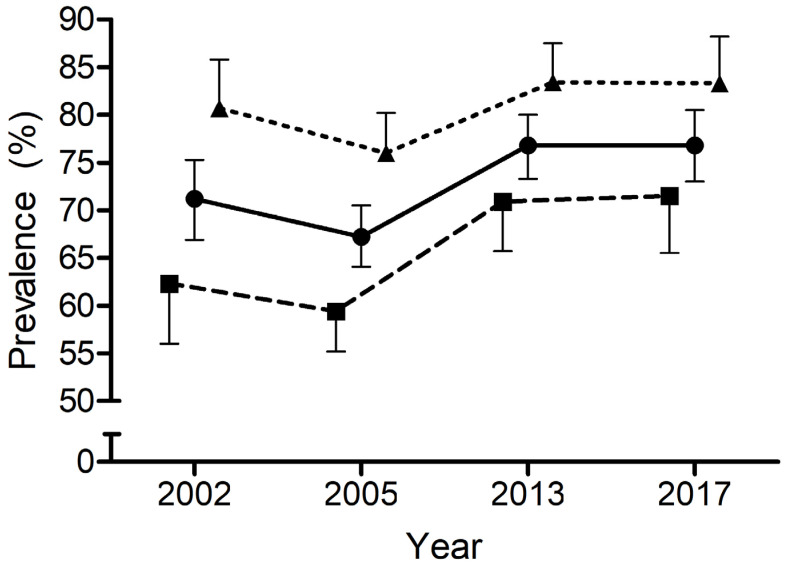
Physical inactivity levels of European Adolescents (boys, girls, and whole sample) in the four studied time points. Levels (%) of physical inactivity (reporting <60 min/day of moderate to vigorous physical activity on average) among European Union Adolescents (in circles, the whole sample; in triangles, the girls’ sample; and in squares, the boys’ sample) for four different time points (2002, 2005, 2013, and 2017). Data are means ± CI.

**Table 1 ijerph-20-03758-t001:** Levels (%) of physical inactivity (PIA) in adolescents (15–17 years old) in the European Union (EU) countries in 2002, 2005, 2013, and 2017.

		2002			2005			2013			2017			2002–2017	
		Sample	PIA (%)	95% CI	Sample	PIA (%)	95% CI	Sample	PIA (%)	95% CI	Sample	PIA (%)	95% CI	χ2	*p*-Value
EU total		462	71.2	66.9–75.3	939	67.2	64.1–70.5	561	76.8	73.3–80.0	452	76.8	73.0–80.5	22,461	<0.001
EU boys		239	62.3	56.0–68.6	498	59.4	55.2–64.0	296	70.9	65.7–76.3	249	71.5	65.5–77.5	16,713	0.001
EU girls		223	80.7	75.5–85.8	441	76.0	71.9–80.2	265	83.4	78.5–87.5	203	83.3	78.0–88.2	7796	0.05
Boys vs. Girls	χ2	18.999			28.979			12.174			8.680				
	*p*-value	<0.001			<0.001			<0.001			0.003				
	Z-Score	4.359			5.383			3.489			2.946				
	PIA levels difference	18.40%			16.60%			12.50%			11.80%				

PIA: physical inactivity levels.

## Data Availability

The raw data is owned by the European Commission and available online (Special Eurobarometer 183–6, December 2002: https://dbk.gesis.org/dbksearch/sdesc2.asp?no=3886&search=58.2&search2=&field=all&field2=all&DB=e&tab=0&notabs=&nf=1&af=&ll=10. Special Eurobarometer 246, December 2005: https://dbk.gesis.org/dbksearch/sdesc2.asp?no=4415&search=64.3&search2=&field=all&field2=&DB=e&tab=0&notabs=&nf=1&af=&ll=10. Special Eurobarometer 412, March 2014: https://dbk.gesis.org/dbksearch/sdesc2.asp?no=5877&search=Physical%20fitness%20and%20exercise&search2=&field=all&field2=&DB=e&tab=0&notabs=&nf=1&af=&ll=10. Special Eurobarometer 472, March 2018: https://dbk.gesis.org/dbksearch/sdesc2.asp?no=6939&search=Physical%20fitness%20and%20exercise&search2=&field=all&field2=&DB=e&tab=0&notabs=&nf=1&af=&ll=10) (accessed on 31 March 2022).

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
