# Peer review of "Physical Inactivity Levels of European Adolescents in 2002, 2005, 2013, and 2017"

_ijerph, 2023, doi:10.3390/ijerph20043758_

Round 1

Reviewer 1 Report

I would like to thank the editors of the IJERPH for the opportunity of reviewing this manuscript. In my opinion, the topic of this study is not new. However, still, I find it a very interesting work because helps to close a gap in knowledge about the physical prevalence of European adolescents using the official data from the European Commission. Also, it helps to press the European Commission to increase the data sample of adolescents in their public opinnion surveys (Eurobarometer).

Despite I find the paper interesting, I think there are some major changes to be addressed before considering this work to be published by the IJERPH.

 Abstract

">60 min/day of moderate to vigorous PA on average" - should be ≥60 min/day (in abstract).

 General comments

- Gender and sex are used interchangeably throughout the manuscript - careful correction and discussion of this should be provided.

- Because of small sample size all EU data were aggregated giving only a continent-level assessment - how is this better, especially with a very small sample, than other published findings?

- I suggest the authors follow the convention of using subheadings to structure the methods section (eg Data source, Physical inactivity, Statistical analysis etc).  This will make it easier for the reader to follow the narrative and tune-in to key pieces of information.

- The authors refer repeatedly to “physical inactivity prevalence” throughout the manuscript. I think it is unnecessary, I recommend the authors to use prevalence when only referring physical activity and otherwise simply say physical activity.

 Introduction

- I think this paper was written before the publication of some latest papers analysing physical activity levels of European children and adolescents. Therefore, I recommend the authors to rewrite the introduction including the latest published works in this fields. For instance:

-      HBSC, Global Matrix 4.0 (Reilly et al. Secular trends in child and adolescent physical activity and sedentary behavior internationally: analyses of data from Active Healthy Kids Global Alliance Global Matrices 1.0-4.0. Journal of Physical Activity and Health, 2022)

- The authors has cited the Global Matrix 3.0 findings.  However, the Global Matrix 4.0 findings has been already published (Aubert et al. Global Matrix 4.0 Physical Activity Report Card Grades for Children and Adolescents: Results and analyses from 57 countries. Journal of Physical Activity and Health, 2022) and include some analyses by sex (Silva et al. Evidence of inequities within and between countries of the Global Matrix 4.0 Physical Activity Report Cards for Children and Adolescents. Journal of Physical Activity and Health, 2022).

- Sawyer, S. M., Azzopardi, P. S., Wickremarathne, D., & Patton, G. C. (2018). The age of adolescence. The Lancet Child & Adolescent Health, 2(3), 223-228.

- Line 37: I think the recommendations are 3 days a week, please, check

- Line 44. WHO is encouraging all member states to achieve the 15% reduction - not just European countries.

Methodology

- The authors truncated responses regarding the amount of time spent in physical activity in an attempt to standardize it across questionnaires. This is
reasonable, but it wouldnot be better trunk the responses the other way around. I.e., categorize responses of “15 minutes” as “30 minutes or less,” as this does not require any assumptions?

Results

Figure 1 and Table 1 present the same data - both are not necessary in the paper - perhaps leave the figure and include the table in supplementary materials?

Discussion and limitations

- A comprehensive discussion of the limitations and inconsistencies of international child and adolescent physical activity surveillance was recently published (Aubert et al. Global prevalence of physical activity for children and adolescents; inconsistencies, research gaps, and recommendations: A narrative review. International Journal of Behavioral Nutrition and Physical Activity 18:81, 2021) - further discussion of these in relation to the data in this paper is needed (e.g. limitations of methods used in Eurobarometer surveys; different methods used in 2002/2005 versus 2013/2015 - similar criticism used to discount the Guthold paper) - merely listing these as limitations in the paper does not justify their use without further expalation/justification.

- The discussion highlights that the present monitoring structure is not capturing health behaviours in children and adolescents in EU with sufficient detail.

Author Response

Thank you very much for your feedback. We have tried our best to addressing the suggested improvements. I respond to all your comments in the attached document

Reviewer 2 Report

Sufficient physical activity is an important factor in the prevention of various diseases. Pointing out the high incidence of inactivity represents an important importance in science and also in clinical practice. I recommend accept publication but I have a few suggestions.

1. It is standardized name Sport and Physical Activity (PA) Special Eurobarometer surveys?
2. More citation is needed “physical inactivity (PIA) has been associated with a negative impact on adolescents’ physical, mental, and social health and addressing this represents a public health priority [1].“
3. You have a lot of information about physical inactivity, but add information about benefit of physical activity a health, please 
4. Add some citation study of Eurobarometer surveys        
5. More information about multi-stage sampling, random design is needed    6. Add STROBE checklist    

Author Response

Thank you very much for your feedback. We have tried our best to addressing the suggested improvements. We respond to all your comments in the attached document

Reviewer 3 Report

Dear Authors,

Review comments are seen below:

Comment 1: In the PDF file, there are some extra space need to be taken care of.

Line 38, ‘[1].  Adolescents......’, it seems like an extra space was left in front of the word ‘Adolescents’.

Line 41, social health  and addressing...,  it seems like an extra space was left in front of the word ‘and’.

Line 45, ....outline  23....., it seems like an extra space was left between these two words. Please check the writing all over if necessary.

......

Comment 2: I understand that there can be always reporting bias when using surveys among different populations, especially conducting surveys among different territories/countries with distinct living behaviors. And from Line 111, the authors demonstrated their method to reduce the bias from approaching different datasets across the year 2002, 2005, 2013, and 2017 surveys, I would suggest the authors to clearly demonstrate all their adaptations across different years of surveys.

Comment 3: ‘for the case of PA, responses of “30 minutes or less” were assumed to mean 15 min; “31 to 60 minutes” was assumed to mean 45 min; “61 to 90 minutes” was assumed to mean 75 min; “91 to 120 minutes” was assumed to mean 105 min; and “more than 120 minutes” was assumed to mean 120 min....’ can generally reflect their true PA, but for studies with small number of sampling participants, this may not always be the case, especially for those who conducted more than 120 min of PA were assumed to mean 120 min, their true PA can be greater than 120 min in some case.’ The authors can make the rounding up more precise if it is appropriate for them.

Comment 4: Any supporting references for Line 189 ‘PIA prevalence of EU adolescents remains high, and no improvements were achieved between 2002 and 2017’? I understand your Table 1, it clearly demonstrated that 2002-2017 PIA% increased from 71.2% to 76.8%, p-value less than 0.001, which can be largely affected by their randomized sampling strategy. Because it can be different when looking at people living in the rural places compared to the urban cities. An explanation/description of what is the percentage of people living in the urban area of the HBSC project Eurobarometer study every trial can make this much clear to the audience. In this case, we would assume the enrolled participants were consisted with equal percentage of people from urban area each year.

Comment 5: I do not see your answer of IRB approval. Please answer Line 283-292, ‘Institutional Review Board Statement: In this section, you should add the Institutional Review Board Statement and approval number, if relevant to your study. You might choose to exclude this statement if the study did not require ethical approval. Please note that the Editorial Office might ask you for further information. Please add “The study was conducted in accordance with the Declaration of Helsinki, and approved by the Institutional Review Board (or Ethics Committee) of NAME OF INSTITUTE (protocol code XXX and date of approval).” for studies involving humans. OR “The animal study protocol was approved by the Institutional Review Board (or Ethics Committee) of NAME OF INSTITUTE (protocol code XXX and date of approval).” for studies involving animals. OR “Ethical review and approval were waived for this study due to REASON (please provide a detailed justification).” OR “Not applicable” for studies not involving humans or animals.’

Reviewer

Author Response

(The authors gave the same response as above.)

Round 2

Reviewer 1 Report

The revised version correctly addresses all aspects of improvement that were pointed out to the researchers. I am happy with its publication in its current form.